# Dimensions of HIV-related stigma in rural communities in Kenya and Uganda at the start of a large HIV 'test and treat' trial

Cecilia Akatukwasa[1]*, Monica Getahun[2], Alison M. El Ayadi[2], Judith Namanya[1‡], Irene Maeri[3‡], Harriet Itiakorit[1‡], Lawrence Owino[3‡], Naomi Sanyu[1‡], Jane Kabami[1‡], Emmanuel Ssemmondo[1‡], Norton Sang[3‡], Dalsone Kwarisiima[1‡], Maya L. Petersen[4‡], Edwin D. Charlebois[5‡], Gabriel Chamie[6‡], Tamara D. Clark[6‡], Craig R. Cohen[2‡], Moses R. Kamya[1,7‡], Elizabeth A. Bukusi[3‡], Diane V. Havlir[6‡], Carol S. Camlin[2,5]

1 Infectious Diseases Research Collaboration, Kampala, Uganda, 2 Department of Obstetrics, Bixby Center for Global Reproductive Health, Gynecology & Reproductive Sciences, University of California, San Francisco, CA, United States of America, 3 Centre for Microbiology Research, Kenya Medical Research Institute, Nairobi, Kenya, 4 Divisions of Biostatistics and Epidemiology, School of Public Health, University of California, Berkeley, Berkeley, CA, United States of America, 5 Center for AIDS Prevention Studies, University of California, San Francisco, San Francisco, CA, United States of America, 6 Division of HIV, Infectious Disease and Global Medicine, Department of Medicine, University of California, San Francisco, CA, United States of America, 7 Makerere University College of Health Sciences, Kampala, Uganda

◉ These authors contributed equally to this work.
‡ These authors also contributed equally to this work.
* cakatukwasa@idrc-uganda.org

**Data Availability Statement:** All data are available within the paper and the interview guides are provided as Supporting Information.

## Abstract

HIV-related stigma is a frequently cited barrier to HIV testing and care engagement. A nuanced understanding of HIV-related stigma is critical for developing stigma-reduction interventions to optimize HIV-related outcomes. This qualitative study documented HIV-related stigma across eight communities in east Africa during the baseline year of a large HIV test-and-treat trial (SEARCH, NCT: 01864603), prior to implementation of widespread community HIV testing campaigns and efforts to link individuals with HIV to care and treatment. Findings revealed experiences of enacted, internalized and anticipated stigma that were highly gendered, and more pronounced in communities with lower HIV prevalence; women, overwhelmingly, both held and were targets of stigmatizing attitudes about HIV. Past experiences with enacted stigma included acts of segregation, verbal discrimination, physical violence, humiliation and rejection. Narratives among women, in particular, revealed acute internalized stigma including feelings of worthlessness, shame, embarrassment, and these resulted in anxiety and depression, including suicidality among a small number of women. Anticipated stigma included fears of marital dissolution, verbal and physical abuse, gossip and public ridicule. Anticipated stigma was especially salient for women who held internalized stigma and who had experienced enacted stigma from their partners. Anticipated stigma led to care avoidance, care-seeking at remote facilities, and hiding of HIV medications. Interventions aimed at reducing individual and community-level forms of stigma may be needed to improve the lives of PLHIV and fully realize the promise of test-and-treat strategies.

**Funding:** Research reported in this article was supported by Division of AIDS, NIAID of the National Institutes of Health (NIH), under award numbers U01AI099959 and UM1AI068636, and in part by the President's Emergency Plan for AIDS Relief (PEPFAR), the Bill and Melinda Gates Foundation, and Gilead Sciences. Gilead Sciences donated tenofovir-emtricitabine (Truvada) in kind. Disclaimer: The content is solely the responsibility of the authors and does not necessarily represent the official views of the NIH, PEPFAR, Bill and Melinda Gates Foundation, or Gilead. None of the sponsors played a role in the study design, data collection, and analysis, interpretation of the data, preparation of the manuscript or the decision to submit the manuscript.

**Competing interests:** Gilead Sciences provided tenofovir-emtricitabine (Truvada) in kind. This does not alter our adherence to PLOS ONE policies on sharing data and materials. The authors do not have any other competing interest to declare.

# Introduction

In Sub-Saharan Africa, persistent HIV-related stigma is an impediment to the success of the global response to HIV [1]. Despite expanded access to antiretroviral therapy (ART), low uptake of HIV-testing among men, delayed engagement in care, and high attrition from care among key populations [2–5] are consequences of HIV-related stigma. Different forms of stigma are implicated as a barrier to achieving the full potential of numerous clinical advances to improve health at the population level, including HIV testing and counselling, voluntary medical male circumcision, pre-exposure prophylaxis (PrEP), ART treatment as prevention, and option B+ for expectant mothers [4, 6–12].

Goffman's conceptualization of three distinct though inter-related forms of enacted, internalized, and perceived stigma [13], as well as subsequent conceptual development in the context of HIV [14] has guided much of the prior research on ways in which stigma has affected people living with HIV (PLWHIV) and impacted upon the success and failures of prevention efforts and interventions to improve health outcomes among PLWHIV. These forms of stigma are manifested along the HIV care continuum. Internalized stigma among PLHIV is known to have resulted in their deferring marriage, sex, and childbearing, distancing from friends and family members, and avoiding health care [15–18].

Perceived stigma, and more specifically, anticipation of HIV-related stigma, has been seen to result in high levels of depression, social isolation and disruptions in normal social relationships, a reluctance to disclose HIV status, and has negatively impacted upon uptake of HIV testing services, care seeking behaviours, and medication adherence among PLWHIV [3, 9, 10, 14, 15, 19–27]. Studies also reveal that anticipated stigma is influenced by age and level of engagement in care and treatment, and that older persons (40 years and over) and those stable on ART are less likely to anticipate stigmatizing attitudes [28].

Enacted stigma, the lived experience of discrimination against a PLWHIV through others' actions or words, is also known to have a negative impact on health outcomes among PLWHIV [29]. In everyday situations, enacted stigma is manifested by repeated acts of gossiping, laughing at and mockery of PLHIV [28].

While some research has described decreases in certain domains of HIV-related stigma over time, largely associated with increasing acceptance of ART and its concomitant positive effects on health as well as intervention efforts targeting stigma reduction [30–33], normative beliefs that HIV infection results from 'sin' or moral failure, and blame and accusations against people living with HIV (PLHIV) for their illness persist in many settings in sub-Saharan Africa [22, 34, 35], and continue to challenge efforts to improve HIV outcomes among PLHIV [5, 6]. Recent literature has shown that misinformation and fears of contagion continues to lead to discrimination and exclusionary acts [22, 29]. Anthropologists have explored cultural underpinnings of the fears and spiritual insecurities [36] that emerged with HIV and contributed to production of HIV-related stigma in Africa, the "regimes of accusation" [37] that have centred HIV/AIDS within concerns about 'promiscuity' and stigmatized expressions of sexuality [34, 38]. These negative attitudes perpetuate stigma and discrimination, impacting the quality of care given to HIV patients [22, 35] and leading to the maltreatment of PLWHIV by their partners, families, and communities, ultimately resulting in poor mental and physical health outcomes [34, 39–41] as well disruption of their HIV care [28].

In the context of the scaling up of universal HIV testing and ART, a deeper understanding of the experiences of persistent stigma among PLWHIV in communities is critical for design and implementation interventions to end the HIV epidemic. This qualitative study sought to explore attitudes towards PLWHIV in the context of the baseline year of a large test-and-treat trial in rural Kenyan and Ugandan communities, in order inform individual and community

stigma reduction interventions in countries undergoing rapid scale up of World Health Organization universal HIV testing and ART access guidelines [42].

## Materials and methods

### Study context

The study was conducted among a sample of participants (age 15 and older) selected from SEARCH (Sustainable East Africa Research in Community Health), a community-level cluster randomized controlled "universal test and treat (UTT)" trial in 32 communities in three regions of Uganda and Kenya exploring the effect of ART initiation at any CD4 count with streamlined delivery [43]. SEARCH demonstrated that its interventions of hybrid mobile testing campaigns [43] and flexible, patient-centered "streamlined care" led to testing, linkage and viral suppression rates that surpassed the 90-90-90 goals after three years [44, 45]. The qualitative study was embedded within 8 of the 32 communities of the trial (Fig 1) to characterize the diverse social and cultural contexts of the SEARCH intervention, including the prevalence of forms of stigma and discrimination towards PLWHIV in the baseline year of the trial.

**Qualitative study sites.** SEARCH rural communities are comprised one or more national geopolitical units just above the village level (i.e., a parish in "Uganda" and "a sub-location" in Kenya) with an average population of 10,000, and located within the catchment area of a President's Emergency plan for AIDS Relief (PEPFAR)-supported government clinic in Southwestern Uganda, Eastern Uganda or western Kenya. 16 matched community pairs were selected based on region, population density, occupation mix, access to transport routes and number of trading centers. Qualitative sub-study sites comprised two randomly selected matched intervention and control communities in southwestern Uganda, two in eastern Uganda, and four in western Kenya [43].

### Qualitative study team

The study co-investigator/qualitative lead led a gender-balanced team of eight US and eastern Africa-based researchers. Team members were trained in qualitative research methods and

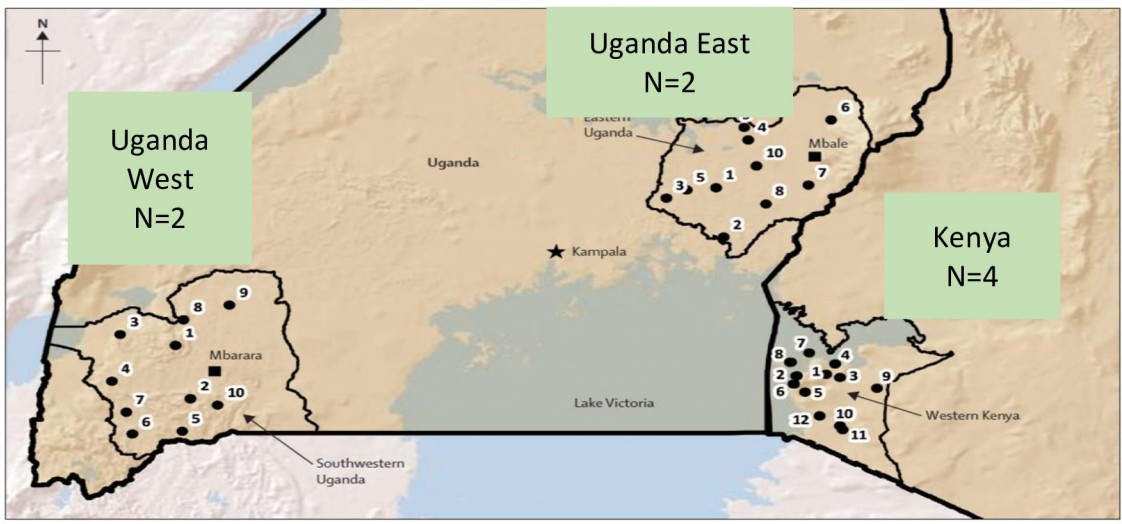

**Fig 1. SEARCH study communities.**

analysis. All team members, including those who collected the data, were involved in the analyses, interpretation, and writing of this manuscript.

## Participants and procedures

The qualitative study utilized both purposive and systematic sampling methods to compose three cohorts of participants: community leaders, HIV care providers, and community members. The cohort of community leaders (n = 32) was selected purposively for gender balance from among the local leaders engaged by SEARCH to help mobilize communities (including members of local councils, Village Health Teams and fishing community leaders, selected for the cohort independent of their HIV status). The cohort of providers (n = 50) was selected purposively to include the cadre of clinical officers, nurses, and HIV counsellors providing care at clinics serving the study communities. To attain heterogeneity in the community cohort by theoretically salient characteristics such as gender and HIV status, the cohort of n = 112 community members was systematically selected (systematic progression through a list of eligible participants organized by HIV status and gender) from a stratified random sub-sample of HIV-positive and -negative community members participating in a household socioeconomic survey in the study communities. To attain heterogeneity in the sample by treatment status at baseline the HIV-positive members the community cohort was further selected to compose equally-sized groups of individuals linked to care but with high CD4 count/unsuppressed viral load, those linked to care with suppressed viral load, and those unlinked to HIV care at baseline. All cohorts sought balance by gender to ensure a range of perspectives. A total of 184 interviews were included within the analyses. Participant characteristics are presented in Table 1.

## Data collection and analysis

Baseline data was collected between February to December 2014. In-depth interviews were conducted utilizing semi-structured guides in the local languages (Ateso, Dholuo, Lugwere, Lusoga, and Runyankole). Privacy and confidentiality were ensured during the data collection process by selecting private, comfortable and convenient spaces for interview. The community member and community leader interviews explored perceived social norms and practices related to HIV testing, disclosure, and care seeking, while provider interviews explored individual and community-level perceptions of HIV and challenges to ART initiation and HIV care provision. All interviews explored individual and community-held beliefs about PLHIV and HIV including stigma and discrimination. Interviews were translated, and transcribed audio recordings into English. An inclusive master code list was developed utilizing the areas of inquiry from the interview guides and a priori codes were defined and applied by an eight-person team on the basis of theory; codes were iteratively refined during the analytical process. Data analyses were conducted using Atlas.ti 7.0. We categorized the diverse experiences of stigma and its consequences in domains of enacted, internalized, and perceived stigma, and explored how these experiences were patterned by regional and other characteristics. We present verbatim quotes to highlight the rich experiences of PLHIV experiencing stigma, as well as community and provider perspectives about persistent stigma.

## Ethical approval

The study received ethical approvals from the University of California San Francisco Committee on Human Research, the Ethical Review Committee of the Kenya Medical Research Institute, the Makerere University School of Medicine Research and Ethics Committee, and the

**Table 1. Participant characteristics.**

| Cohort Type | Characteristic | N (%) |
|---|---|---|
| **Community Cohort** | **Sex** | |
| **N = 112** | Female | 65 (58.0%) |
| | Male | 47 (42.0%) |
| | **Age** | |
| | Median | 38 years |
| | **Serostatus** | |
| | Positive | 70 (62.5%) |
| | Negative | 42 (37.5%) |
| | **Region** | |
| | Kenya | 56 (50%) |
| | Uganda East | 28 (25%) |
| | Uganda West | 28 (25%) |
| **Community Leader** | **Sex** | |
| **N = 32** | Female | 12 (37.5%) |
| | Male | 20 (62.5%) |
| | **Age** | |
| | Median | 50 |
| | Range | (14–72) |
| | **Region** | |
| | Kenya | 16 (50.0%) |
| | Uganda East | 8 (25.0%) |
| | Uganda West | 8 (25.0%) |
| **Health care workers** | **Sex** | |
| **N = 50** | Female | 27 (54.0%) |
| | Male | 23 (46.0%) |
| | **Region** | |
| | Kenya | 28 (56.0%) |
| | Uganda East | 10 (20.0%) |
| | Uganda West | 12 (24.0%) |

*Missing age for 8 community leaders: age was not asked for all cohorts same as sero-status.

Uganda National Council for Science and Technology. All study participants provided written confirmation of informed consent to participate in the study during enrolment.

## Results

Participants shared a range of beliefs, attitudes and experiences with the dimensions of HIV-related stigma, and findings revealed differences across geographic contexts and underlying HIV prevalence levels of communities in the first year of the SEARCH trial. Narratives provide evidence for stigmatizing attitudes, including blaming PLWHIV for their infection, and associating a diagnosis of HIV with moral decay or 'promiscuity'. The discourse in many communities included discussions about HIV as an outcome of 'avoidable' and 'sinful' behavior including 'prostitution', promiscuity, or adultery. These views were more common in communities of low HIV prevalence, specifically in western and eastern Uganda, and were more often found among narratives of older community members and women.

'*I think it is brought about by prostitution and promiscuity, do you think that if you are not disorganized you get that illness [HIV]? It is a disease that you bring upon yourself; it is not like measles that you just get*' (HIV-negative female, 80 years old, Uganda)

'*It [HIV] came out of promiscuity because I know when you say that 'so and so has HIV, they say... Eh! All this time she has been a prostitute, where did she get it from [...] because it came from the sin of promiscuity*' (older HIV-negative female, Uganda)

'*More often than not people think that those who are HIV infected are promiscuous. They say yes to any sexual advancement...*' (HIV-positive male, Kenya)

'*When people get to know that you have HIV, most of them start treating you as someone who has no brains. They start talking about how you got involved in adultery*' (HIV-positive male, Uganda)

### Experiences of enacted stigma

Enacted stigma was expressed as verbal discrimination and humiliation, and rejection, including explicit acts of segregation, which were largely motivated by fear of contagion. It included both personal experiences of discrimination by PLHIV, and discriminatory acts and attitudes towards PLHIV as reported by HIV-negative participants. Verbal abuse was the most frequently reported enacted stigma. Mocking, harassment, pejorative labelling and name calling was frequently used for humiliation and public devaluation of PLHIV, including referring to them as close to death or already dead, as illustrated by the following quotes:

'*They say that "I am sure she is about to die; she is just a moving corpse". When you hear that you start to feel bad.*' (HIV-positive female, Uganda)

'*I have heard people calling them derogatory names like 'Otolo' to mean those who have benefited from use of ARVs [....] In this beach, if you are not a strong willed fisherman, then you cannot survive*' (HIV-positive male, Kenya)

'*Some community people like to laugh at us HIV positive people. They say "that person is about to die; dead bodies! You are dead people!"*' (HIV-positive female, Uganda)

'*People laugh at us... they point fingers saying that we are dead [...] when people are in groups they talk ill about people with HIV saying that those ones [PLHIV] are finished [....] One man commented, "our village is going to become Luwero [Luwero is a district in Uganda which was heavily impacted by HIV and the civil war in the 1980s]" meaning that all people in the village are going to die off*' (HIV-positive female, Uganda)

'*You hear people say that "X died a long time ago", and you feel like your life is not good and you hate everything*' (HIV-positive female, Uganda)

Similar perceptions of HIV as a death sentence were reflected in comments by HIV-negative community members, evidenced in the following quotation:

'*It is because I know that being tested HIV positive means that I am a death candidate [...] Thank God I am HIV negative.*' (HIV-Negative male, Uganda)

Participant narratives also demonstrated significant challenges in the home environment related to intimate relationships, including loss of support, outright rejection, and physical

violence; these findings were exclusive to women, no partner rejection or abuse was reported as having occurred to men.

> '*There was a pregnant woman who tested HIV positive and was advised to disclose her status to the husband. The husband was not amused. . .he rejected her and she died two days later out of shock in her 7 months of pregnancy.*' (HIV-negative female, Kenya)

> '*There was even another woman whose husband refused to access HIV care till they separated [. . .] the husband even traced her up to Migori District Hospital (about 20 km away) where she opted to access HIV care. He waited for her by the gate and beat her thoroughly and even snatched her drugs.*' (Male provider, Kenya)

Several respondents shared how PLHIV were physically excluded from participating in social activities or how physical contact with PLHIV was avoided, even by way of inanimate objects, largely stemming from irrational fears of contamination and infection among HIV-negative community members:

> '*The people that do not have HIV do not talk well about those that have it [PLHIV]; they abuse them [. . . .] For example, if that HIV-positive person is going to touch something, they tell him, "do not touch it because you died a long time ago!"*' (HIV-Positive female, Uganda)

> '*People don't want to use a cup that an HIV patient has used and they throw away razor blades or safety pins that the patient touches because they don't want to contract HIV. Some even tell you not to touch their belongings because you are sick.*' (HIV-positive female, Uganda)

One HIV-negative respondent also voiced a preference for having quarantined HIV-positive individuals to prevent the spread of HIV:

> '*I feel the government ought to have done something when HIV started. They [PLHIV] ought to have been isolated from the non HIV infected persons, somewhere that they just stay on their own and being fed well to avoid further transmission*' (HIV-negative male, Kenya)

Other areas of exclusion of PLHIV were evident in narratives around community leadership and livelihood opportunities:

> '*There is open discrimination and name calling that demoralizes the infected persons. Some even tell you on the face that you cannot do certain jobs that require strength because you are on drugs (ARVs).*' (HIV-negative male, Kenya)

> '*He [HIV-positive patient] gave us a testimony that if you are sick and you talk, especially government workers, you get problems because everyone sees you as someone not alive, not managing and worthless.*' (Female provider, Uganda)

> '*Like political and leadership opportunities in the community, people will not be willing to give someone [PLHIV] responsibility who is going to die soon.*' (HIV-negative male, Kenya)

## Internalized stigma and its consequences

Narratives revealed an internalization of social attitudes, fears, and discrimination towards PLHIV. Evidence of internalized stigma was found in participant accounts of feelings of

worthlessness, shame, and embarrassment. Participants expressed difficulty accepting their serostatus, embarrassment associated with their HIV infection, and shame, which made it difficult to not only disclose their HIV status, but to also interact with others. Narratives also described depressive symptoms and suicidal thoughts among PLHIV. These accounts reveal the emotional distress brought on by internalized stigma and the distancing of self that occurs in self-stigmatizing individuals.

PLHIV evidenced internalized stigma in discussions about feeling worthless, non-existent, and being 'dead'. Others describe shame or embarrassment about their HIV-status, often associated with the community perception that HIV is a disease affecting the immoral or promiscuous. Fear, shame, and embarrassment were common responses to others' awareness of a participant's HIV-positive status:

'*I was feeling ashamed and feared others disclosing my status to the public.' (HIV-positive female, Kenya)*

'*When someone knows that you are sick, you have Siriimu [HIV], and you get embarrassed.' (HIV-positive female, Uganda)*

'*At times you hear a person say in public that you have HIV and you start to feel embarrassed [. . .] you see this disease does not [have a] cure and everyone thinks it is the worst disease; in fact, the people in my village think that way.' (HIV-positive female, Uganda)*

A woman discusses embarrassment as a reason for not disclosing her HIV status to her immediate family:

'*My mother has died but I can't tell my relatives. They can only know at the time when I am bedridden but if I am still looking healthy and doing my duties as normal, I wouldn't want them to know [. . .] I feel embarrassed [. . .] what will they think of me, that our sibling is on ARVs.' (HIV-positive female, Kenya)*

HIV-positive individuals' acute awareness and internalization of the negative community perceptions about HIV led to significant mental health consequences and distancing from family members and friends. Narratives revealed extreme cases of anxiety and depression marked by suicidal thoughts and attempts, revealed by HIV-positive individuals, and by their HIV-negative friends or healthcare providers. Many of these situations were precipitated by an HIV-positive diagnosis or a disclosure event.

'*[There was] not even a friend could I meet or talk to at that time. I even failed to move to the public. I was all the time in bed. Time came when I felt like committing suicide' (HIV-positive male, Uganda)*

'*On telling her the results she refused and even reacted badly about the news that she was HIV positive. She had a lot of anxiety and depression and it seems at one time she wanted to commit suicide' (Male provider, Uganda)*

One woman described the failed suicide attempt of her husband and co-wife, upon her discovery of their HIV-positive status:

'*My co-wife used to hide the drugs until a neighbour leaked this out to me. When I told my husband [. . .], he opted to commit suicide together with my co-wife and had it not been for a certain man who found them in the bush where they were prepared to die, they could be no*

*more. They were caught offside having mixed the poison and were ready to drink'* (HIV-positive female, Kenya)

## Anticipated stigma and its consequences

Participant report of anticipated stigma included fears of negative attitudes and behaviors towards PLHIV, including judgement, marital dissolution, verbal and physical abuse. These stigma anticipations acted as barriers to care engagement, often leading PLHIV to delay disclosure, hide medication, and seek treatment in remote facilities. Narratives showed that PLHIV concealed their HIV status from their spouses, immediate family members, and the community, because of fears of being judged as forlorn and 'dead':

'*I do not know if every old person thinks like that or it is [my mother] alone [. . .] if I go home today and tell [her], "mother as you see me here I have HIV", from that day she keeps seeing me like a dead person. She has nine children but she will keep counting eight even when I am still alive'* (HIV-positive male, Uganda)

'*That is our culture, when people get to know [that you are positive], you get ashamed, and they see you as a problem [. . .], that you did something outrageous [. . .]. Some people start avoiding you, as if you have no use and they look to you as someone who is dead already'* (Female provider, Uganda)

A disclosure of HIV-positive status was also expected to culminate in social rejection and compromised intimate relationships. This was particularly salient among women, who were often afraid of losing their marriages, and youth, who were anxious about diminished sexual or marital prospects:

'*Some [HIV-positive women] are still sexually active and they do not want to be discovered [. . .] they are in fear that they will be rejected by those who are negative'* (HIV-negative male, Uganda)

'*This is common among the youth and young women [. . .] they would not want to be associated with the disease lest they miss on girlfriends and men lovers respectively'* (HIV-negative male, Kenya)

PWLHA also anticipated public ridicule, gossip and verbal insults from community members:

'*People start to spread rumours about you and you cease to fit in public such that wherever you go people are gossiping about you that you have HIV and I may not feel good about that'* (HIV-positive male, Uganda)

Providers reported how several HIV-positive men controlled their wives' access to HIV care, either through preventing access to ARVs, restricting their clinic attendance, or limiting their access overall, due to fears of their own HIV status being disclosed:

'*There is a lady [that we initiated] on ART. [. . .] The husband asked her why she was always going to the health facility every month [. . ..] The husband did not want her to be seen taking HIV medication and she did not know what to do.'* (Female provider, Uganda)

'*[One female patient] said that her husband does not want people to know that [they have HIV], so they have to come for their drugs at 6:00 pm when everyone else has left the clinic.'* (Female provider, Uganda)

'*I once heard from a friend [. . .] that her husband was throwing away her drugs and some other hospital supplies meant for those who are living with HIV. He was doing this not knowing that he was already infected as well.*' (HIV-positive female, Kenya)

## Strategies for coping with or avoiding anticipated stigma

Anticipated stigma had a significant impact on care seeking and HIV medication adherence. Participants shared a number of coping strategies to avoid the anticipated stigma of inadvertent disclosure of HIV-positive status while seeking HIV care, including avoiding care altogether, modifying the location where they sought care, avoiding seeking care during peak hours, and not associating with services or incentives reserved for HIV-positive patients. A male community member discusses avoiding care as a result of shame:

'*I was saying people will say 'oh so and so is having HIV' so I kept it to myself until I went down and down until I was bedridden.*' (HIV-positive male, Kenya)

Providers describe patients avoiding care in an effort to avoid the stigma associated with being seen by someone he/she knows, including health workers, whom they did not trust to maintain privacy:

'*Some clients fear other health workers at the facility. We one-time had a scenario of a client who denied being in care after seeing a health worker whom the client knew. We later realized that it was a stigma issue.*' (Female provider, Uganda)

'*She said that if she came to the clinic and met that health worker, the whole village would get to know about her status because the health worker was her relative and that he would tell everyone about her condition.*' (Male provider, Uganda)

Providers describe care seeking in remote facilities as a coping strategy for anticipated stigma:

'*Because of stigma [. . .] the person opts to even walk to Mbita district hospital (6km) or even to Ogongo sub district hospital (31km) simply because of stigma.*' (Male provider, Kenya)

'*They [patients] continue trying all efforts to get treatment from far where he is not known. So the biggest thing is that they are ashamed.*' (Male provider, Uganda)

A provider describes patients' practice of care seeking outside of normal clinic hours as a strategy to avoid anticipated stigma:

'*When other patients come at the normal time the other patients with the problem of stigma tend to come late in the evening and we have to help them even when they come late which we find challenging.*' (Male provider, Uganda)

PLHIV seeking care at the clinic often avoided being near rooms or areas designated for PLHIV; waiting in these areas was considered to be an inadvertent disclosure:

'*She told me that she can't sit in that tent at the facility [. . .] She said that people will see her coming to the clinic and they will know that she is sick [. . .] I waited for her the next day [. . . but] she has not showed up yet.*' (Male provider, Uganda)

Patients also refused clinic-offered incentives reserved for PLHIV for fear of association with HIV.

'*There were jerry cans we were given as the HIV-positive people but in this whole village people feared to pick them with the exception of my family just because they fear to be identified as being HIV[-positive]*' (HIV-positive female, Uganda)

In some cases, anticipated stigma associated with being seen in possession of ARVs (both the packaging and the actual pills) led participants to hide their drugs, and resulted in difficulty adhering to ARTs. A provider describes patients' strategies to avoid stigma:

'*The [female patients] used to throw the [ART medication'] boxes in the bushes. But these days they are used; they just leave the boxes in the bins [throw away the drugs at the health facility]. So it is just stigma and the way they stay in their particular homes. The [husbands] do not allow them to take the drugs*' (Male provider, Kenya)]

## Discussion

This qualitative study demonstrates the persistence of HIV-related stigma in a setting that had experienced the HIV epidemic for over 30 years prior to the start of a large universal HIV testing and treatment trial in 2014. At the time the SEARCH study began to be implemented, participants reported that overt cases of enacted stigma had reduced dramatically since the early days of the epidemic, which is supported by prior literature [6, 46], yet felt that gossiping, mockery, distancing and fear of contagion persisted. Further, enacted stigma had evolved into more subtle discriminatory acts/views, including exclusion of PLHIV from leadership and livelihood/work opportunities, as reported in other studies [47].

Internalized stigma narratives revealed intense feelings of shame and worthlessness, resulting in self-isolation and several accounts of suicidal thoughts and attempts. These narratives were often described as reactions or thoughts following HIV-diagnosis among individuals who had not yet come to terms with their diagnosis. The instances of internalized stigma in our findings were fueled by cultural and religious norms of communities and the association of HIV with promiscuity. As is the case in other studies [34], this internalized stigma appeared to be intensely-felt and tied to value judgments and deeply held beliefs about PLHIV, specifically the belief that HIV-positive individuals were less than, damaged, and tainted [48]. Internal stigma was especially salient if PLHIV lacked supportive spouses and family members who accepted their HIV-positive status. Internalized-stigma also appeared to be more intense and frequently reported among women, especially as it related to promiscuity and sex- i.e. the belief that only the promiscuous were infected with HIV. These internalized feelings often resulted in concealment of HIV status from family members and friends. Our findings support prior qualitative research finding that PLWHA expressed anxiety about partner abandonment, relationship dissolution or violent responses if their partners learnt of their HIV status, compromising support they receive from their partners [49, 50]. Similarly, our findings match prior work showing that persons with a greater degree of internalized stigma have been less likely to disclose their sero-positivity to social ties of all types including sexual partners [51], and that internalized stigma also has resulted in anxieties among PLWHIV which deter them from pursuing their fertility goals and marital aspirations [15–18].

In these narratives, anticipated stigma had had a significant impact on PLWHIV's willingness to engage and remain in care, prior to the start of the SEARCH study. Participants reported that anticipated stigma had compromised their participation in HIV testing, linkage to HIV care, and ART adherence (in the past and in the first year of SEARCH), as has been reported in prior studies [9, 10, 17, 21, 27]. Our study revealed that the prevalent moral discourse that HIV is a result of prostitution or adultery resulted in stigma anticipations including public ridicule, gossip or social exclusion. Anticipated stigma was a significant impediment to

HIV care engagement, particularly for women whose own stigma anticipations interacted with experiences of enacted stigma as a result of partner stigma anticipations (i.e., their partners limited their access to HIV-related care for fear of being associated with someone who is HIV-positive). This finding is consistent with prior research finding that uptake of HIV testing services among women was limited by anticipation of community level stigma; while for men, it was due to individually-held anticipated stigma [52].

Finally, our findings also reveal that efforts to access HIV care were hindered by PLWHIV's concealment strategies to avoid the anticipated stigma and inadvertent HIV status disclosure thought to be associated with care seeking. Other studies have shown patients' adopting these concealment behaviours because of stigma anticipations [10, 31, 53]. In several narratives, even if individuals were able to overcome their anticipated stigma and seek care, adherence and care retention remained a challenge. Similar findings have also been reported in settings where scale-up of ART is taking place [6, 51].

Our findings emphasizing the important role of social and gender-specific norms and power structures in influencing local constructions and experiences of HIV-related stigma within this setting underscore the need to closely focus on broad social and structural factors within the development and implementation of stigma reducing interventions [14, 54]. Such an approach would include the adoption of an intersectional approach to acknowledge and address variation in individuals' stigma experiences based on the multiple stigmatized identities they may inhabit and the interaction between these [55].

This study was subject to limitations. Data were drawn from a qualitative study conducted to characterize the diverse social and cultural contexts of the SEARCH trial during the baseline year of the trial. The questions used to gather the data focused on community attitudes towards HIV/AIDS rather than first-hand accounts and experiences of stigma. However, our data sources were inclusive of a broad range of perspectives and data collection methods. The study utilized both in-depth interviews and focus group discussions to surface community norms as well as individual experiences of stigma that may not have surfaced in group discussions. Additionally, there was strong concordance among different informants (i.e. findings were supported by PLWHIV, providers, and HIV-negative community members). Some of the findings were also observed by the study team during the data collection process (e.g., fear of inadvertent disclosure or association with an HIV-positive status). To strengthen the findings, all team members, including data collectors, were involved in the coding, analysis, and interpretation of findings. The study was conducted in two different regions of Uganda (eastern and south-western Uganda) as well as in-land and island western Kenya communities, which represent varied sociocultural settings. Finally, this study relied on cross-sectional analysis of baseline data and therefore did not explore changes in forms of stigma in these communities over time. Recent research in the study setting involving analysis of data collected at multiple time points has suggested changes in stigma did occur, suggesting pathways through which stigma can decline in the context of rapid uptake of HIV testing and treatment [33, 56].

In conclusion, qualitative data collected in the baseline year of a large HIV test and treat intervention trial showed that in rural communities in Uganda and Kenya, HIV-related stigma in all its forms often severely impacted upon the lives of PLHIV, and influenced HIV care seeking behaviors and outcomes among PLHIV at that time. The evidence in this study further indicates that while access to ARTs may have resulted in a reduction of overt cases of enacted stigma and discrimination compared to earlier times in the epidemic, PLWHIV continued to experience internalized and anticipated stigma due to active discourses surrounding HIV and sexuality, assigning moral judgment and blame to PLWHIV, with consequences particularly acute for HIV-positive women. A nuanced understanding of dimensions of stigma may be useful for understanding how HIV-related stigma can affect the implementation of HIV

interventions. Specifically, test-and-treat strategies may be strengthened by a focus on addressing gender inequities and building of positive self-identities among PLWHIV.

## Supporting information

**S1 File.**
(DOCX)

**S2 File.**
(DOCX)

## Acknowledgments

The SEARCH project gratefully acknowledges the Ministries of Health of Uganda and Kenya, our research team, collaborators and advisory boards, and especially all communities and participants involved.

## Author Contributions

**Conceptualization:** Maya L. Petersen, Edwin D. Charlebois, Gabriel Chamie, Tamara D. Clark, Craig R. Cohen, Moses R. Kamya, Elizabeth A. Bukusi, Diane V. Havlir, Carol S. Camlin.

**Data curation:** Monica Getahun, Carol S. Camlin.

**Formal analysis:** Cecilia Akatukwasa, Monica Getahun, Alison M. El Ayadi, Carol S. Camlin.

**Funding acquisition:** Moses R. Kamya, Elizabeth A. Bukusi, Diane V. Havlir, Carol S. Camlin.

**Investigation:** Cecilia Akatukwasa, Judith Namanya, Irene Maeri, Harriet Itiakorit, Lawrence Owino, Naomi Sanyu.

**Methodology:** Maya L. Petersen, Edwin D. Charlebois, Gabriel Chamie, Tamara D. Clark, Craig R. Cohen, Moses R. Kamya, Elizabeth A. Bukusi, Diane V. Havlir, Carol S. Camlin.

**Project administration:** Cecilia Akatukwasa, Monica Getahun, Judith Namanya, Irene Maeri, Harriet Itiakorit, Lawrence Owino, Jane Kabami, Emmanuel Ssemmondo, Norton Sang, Dalsone Kwarisiima, Edwin D. Charlebois, Carol S. Camlin.

**Resources:** Monica Getahun, Dalsone Kwarisiima, Moses R. Kamya, Elizabeth A. Bukusi, Diane V. Havlir.

**Supervision:** Monica Getahun, Dalsone Kwarisiima.

**Validation:** Carol S. Camlin.

**Visualization:** Cecilia Akatukwasa, Monica Getahun, Alison M. El Ayadi, Carol S. Camlin.

**Writing – original draft:** Cecilia Akatukwasa, Monica Getahun, Alison M. El Ayadi, Judith Namanya, Irene Maeri, Harriet Itiakorit, Lawrence Owino, Naomi Sanyu, Carol S. Camlin.

**Writing – review & editing:** Cecilia Akatukwasa, Monica Getahun, Alison M. El Ayadi, Carol S. Camlin.

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
