## [Decision Letter · Decision Letter 0]

21 Oct 2020

PONE-D-20-20464

Dimensions of HIV-related stigma in rural communities in Kenya and Uganda at the start of a large HIV ‘test and treat’ trial

PLOS ONE

Dear Dr. Akatukwasa,

Thank you for submitting your manuscript to PLOS ONE. After careful consideration, we feel that it has merit but does not fully meet PLOS ONE’s publication criteria as it currently stands. Therefore, we invite you to submit a revised version of the manuscript that addresses the points raised during the review process.

We look forward to receiving your revised manuscript.

Kind regards,

Matt A Price

Academic Editor

PLOS ONE

Journal Requirements:

2. When reporting the results of qualitative research, we suggest consulting the COREQ guidelines: http://intqhc.oxfordjournals.org/content/19/6/349. In this case, please consider including more information on the number of interviewers, their training and characteristics; criteria for participants selection; and the dates on which the research was carried out.

3. Please provide additional details regarding participant consent. In the ethics statement in the Methods and online submission information, please ensure that you have specified (1) whether consent was informed and (2) what type you obtained (for instance, written or verbal). If your study included minors, state whether you obtained consent from parents or guardians. If the need for consent was waived by the ethics committee, please include this information.

4. Thank you for stating the following in the Competing Interests/Financial Disclosure * (delete as necessary) section:

"Research reported in this article was supported by Division of AIDS, NIAID of the National Institutes of Health (NIH), under award numbers U01AI099959 and UM1AI068636, and in part by the President’s Emergency Plan for AIDS Relief (PEPFAR), the Bill and Melinda Gates Foundation, and Gilead Sciences. Disclaimer: The content is solely the responsibility of the authors and does not necessarily represent the official views of the NIH, PEPFAR, Bill and Melinda Gates Foundation, or Gilead."

We note that you received funding from a commercial source: Gilead Sciences.

Reviewers' comments:

Reviewer's Responses to Questions

**Comments to the Author**

1. Is the manuscript technically sound, and do the data support the conclusions?

Reviewer #1: Partly

Reviewer #2: Yes

2. Has the statistical analysis been performed appropriately and rigorously? 

Reviewer #1: N/A

Reviewer #2: N/A

3. Have the authors made all data underlying the findings in their manuscript fully available?

Reviewer #1: Yes

Reviewer #2: Yes

4. Is the manuscript presented in an intelligible fashion and written in standard English?

Reviewer #1: Yes

Reviewer #2: Yes

5. Review Comments to the Author

Reviewer #1: This is an interesting and largely well written paper reported on findings from a qualitative study embedded in a cluster randomized trial in “32 communities in east Africa exploring the effect of ART initiation at any CD4 count with streamlined delivery”. The qualitative study aimed to “explore attitudes towards PLWHIV…in order inform individual and community stigma reduction interventions in regions undergoing universal HIV testing and rapid ART scale-up”.

Major comments

1. At the moment, there is insufficient information given about the study methods (sampling and data collection) to have a good understanding of what was done and why, and therefore to have confidence in the findings. This is a large participant group – its looks like a total of 184 individuals (health providers, community leaders and community members) were included, although this number is not directly stated. There is also very little information given about the participants, for example it’s not clear what kinds of ‘community leaders’ were involved, or what the socioeconomic and demographic background of the participants is, or how many were HIV positive. I imagine some of the community leaders and health workers might have been HIV positive, but this has not been explored. Importantly, limited contextual information is provided across the communities involved. In fact, there is no description of what is meant by a community – it seems likely these are geographic communities – or how the 8 communities were selected for inclusion. All of these areas should be explained in more depth – particularly for a qualitative study - including more contextual information about the communities, such as their approximate size as well as their socioeconomic and cultural features. The only supplementary materials I accessed were data collection tools. I suggest more information on the participants, participant groups and study site context should be included, and a table of participants developed to give more detail to this account.

2. It’s also not clear why the team decided to use a random sampling method to identify the community members included in this study, given its qualitative nature. This seems to mix underlying paradigms for qualitative and quantitative research in a very confusing way. I suspect this may have been a practical choice – given the need to sample within a large number of (potentially) large communities – but in the end it’s not at all clear how they ‘narrowed down’ to the final 112 individuals across 8 communities included.

3. The discussion mainly draws on the literature to identify and discuss enacted, internalized and anticipated forms of stigma in the data. I particularly found ideas around anticipated stigma helpful and clearly related to the findings. Given that the authors particularly note the Parker and Aggleton paper (2003), I was surprised not to read more about structural level influences within the discussion, around issues of power and existing inequalities. Gender is clearly pulled out, which is important, but the literature on stigma strnngly underlines the importance of considering wider interrelated structural influences in developing interventions or programmes on HIV stigma, so that the discussion seems to be missing an important element, given the underlying aims of the study.

Minor comments

1. Some sentences in the methods description are hard to understand. For example, on P3, its written that “HIV positive participants were further systematically selected from each community on the basis of randomly generated lists of adults with the following characteristics: high CD4 cell (n=3), suppressed viral load (n=3), and unlinked to HIV care (n=3)”. I suspect it would help to read the main paper for the SEARCH study to understand these descriptions, but it also seems important that this qualitative study should ‘standalone’.

2. In the discussion (P12), the authors write “Similar to other literature, our findings support a dramatic reduction in overt cases of enacted stigma since the early days of the epidemic” – but no findings are presented to support this point, as far as I can see.

Reviewer #2: This sub-study is well justified and findings are useful in guiding the main study.

1. Regarding ethical considerations, authors only discussed REC approval. Was written verbal consent sought from all respondents?

2. Authors briefly need to describe the tool that they used for this sub study.

3. Were there any other ethical issues that the researchers had to address e.g confidentiality issues?

6. PLOS authors have the option to publish the peer review history of their article (what does this mean?). If published, this will include your full peer review and any attached files.

Reviewer #1: No

Reviewer #2: **Yes: **Paul Ndebele

---

## [Author Response · Author response to Decision Letter 0]

15 Mar 2021

Reviewer#1: 

1. At the moment, there is insufficient information given about the study methods (sampling and data collection) to have a good understanding of what was done and why, and therefore to have confidence in the findings. This is a large participant group – its looks like a total of 184 individuals (health providers, community leaders and community members) were included, although this number is not directly stated. There is also very little information given about the participants, for example it’s not clear what kinds of ‘community leaders’ were involved, or what the socioeconomic and demographic background of the participants is, or how many were HIV positive. I imagine some of the community leaders and health workers might have been HIV positive, but this has not been explored. Importantly, limited contextual information is provided across the communities involved. In fact, there is no description of what is meant by a community – it seems likely these are geographic communities – or how the 8 communities were selected for inclusion. All of these areas should be explained in more depth – particularly for a qualitative study - including more contextual information about the communities, such as their approximate size as well as their socioeconomic and cultural features. The only supplementary materials I accessed were data collection tools. I suggest more information on the participants, participant groups and study site context should be included, and a table of participants developed to give more detail to this account.

It’s also not clear why the team decided to use a random sampling method to identify the community members included in this study, given its qualitative nature. This seems to mix underlying paradigms for qualitative and quantitative research in a very confusing way. I suspect this may have been a practical choice – given the need to sample within a large number of (potentially) large communities – but in the end it’s not at all clear how they ‘narrowed down’ to the final 112 individuals across 8 communities included.

Response: 

We appreciate the reviewer’s comments and have now included a table describing the participant characteristics by cohort (Table 1) page 4. We have also revised our description of our sampling approach, which involved both traditional qualitative purposive sampling approaches as well as a systematic sampling approach to obtain a heterogeneous community cohort, which indeed was a practical strategy to compose a cohort with the range of salient characteristics we wanted to be sure to include. The section now reads as follows: 

The qualitative study utilized both purposive and systematic sampling methods to compose three cohorts of participants: community leaders, HIV care providers, and community members. The cohort of community leaders (n=32) was selected purposively for gender balance from among the local leaders engaged by SEARCH to help mobilize communities (including members of local councils, Village Health Teams and fishing community leaders, selected for the cohort independent of their HIV status). The cohort of providers (n=50) was selected purposively to include the cadre of clinical officers, nurses, and HIV counsellors providing care at clinics serving the study communities. To attain heterogeneity in the community cohort by theoretically salient characteristics such as gender and HIV status, the cohort of n=112 community members were systematically selected (systematic progression through a list of eligible participants organized by HIV status and gender) from a stratified random sub-sample of HIV-positive and -negative community members participating in a household socioeconomic survey in the study communities. To attain heterogeneity in the sample by treatment status at baseline the HIV-positive members the community cohort were further selected to compose equally-sized groups of individuals linked to care but with high CD4 count/unsuppressed viral load, those linked to care with suppressed viral load, and those unlinked to HIV care at baseline. All cohorts sought balance by gender to ensure a range of perspectives. A total of 184 interviews were included within the analyses. Participant characteristics are presented in Table 1. 

2. The discussion mainly draws on the literature to identify and discuss enacted, internalized and anticipated forms of stigma in the data. I particularly found ideas around anticipated stigma helpful and clearly related to the findings. Given that the authors particularly note the Parker and Aggleton paper (2003), I was surprised not to read more about structural level influences within the discussion, around issues of power and existing inequalities. Gender is clearly pulled out, which is important, but the literature on stigma strongly underlines the importance of considering wider interrelated structural influences in developing interventions or programmes on HIV stigma, so that the discussion seems to be missing an important element, given the underlying aims of the study.

Response:

Thank you for this recommendation; we now include text about the implications of our findings for addressing structural factors underlying the production of stigma (see page 6). 

Minor comments

3. Some sentences in the methods description are hard to understand. For example, on P3, its written that “HIV positive participants were further systematically selected from each community on the basis of randomly generated lists of adults with the following characteristics: high CD4 cell (n=3), suppressed viral load (n=3), and unlinked to HIV care (n=3)”. I suspect it would help to read the main paper for the SEARCH study to understand these descriptions, but it also seems important that this qualitative study should ‘standalone’.

Response:

As noted above, we have revised the methods section on sampling to improve clarity of the participant categories Page 4

4. In the discussion (P12), the authors write “Similar to other literature, our findings support a dramatic reduction in overt cases of enacted stigma since the early days of the epidemic” – but no findings are presented to support this point, as far as I can see.

Response: 

We now clarify that participants themselves reported that overt cases of enacted stigma had declined, compared to past years, earlier in the epidemic in these communities.

Reviewer #2: This sub-study is well justified and findings are useful in guiding the main study.

5. Regarding ethical considerations, authors only discussed REC approval. Was written verbal consent sought from all respondents?

Response:

We have edited the ethical statement to include how informed consent was obtained; that it was written and voluntary informed consent. We also describe how confidentiality for the study participants was ensured. The study does not include minors, those stipulated in the age category 15-18 years were emancipated minors and therefore no need for assent and consent from parents or guardians. 

6. Authors briefly need to describe the tool that they used for this sub study.

Response: 

We have described our study instruments on page 6 and have attached them as supplemental material. 

7. Were there any other ethical issues that the researchers had to address e.g. confidentiality issues?

Response: 

We provide details regarding how the study handled confidentiality concerns within the data collection and analysis section. (page 5)

---

## [Editor Report · Decision Letter 1]

19 Mar 2021

Dimensions of HIV-related stigma in rural communities in Kenya and Uganda at the start of a large HIV ‘test and treat’ trial

PONE-D-20-20464R1

Dear Dr. Akatukwasa,

We’re pleased to inform you that your manuscript has been judged scientifically suitable for publication and will be formally accepted for publication once it meets all outstanding technical requirements.

Kind regards,

Matt A Price

Academic Editor

PLOS ONE
---

## [Editor Report · Acceptance letter]

7 May 2021

PONE-D-20-20464R1 

Dimensions of HIV-related stigma in rural communities in Kenya and Uganda at the start of a large HIV ‘test and treat’ trial. 

Dear Dr. Akatukwasa:

I'm pleased to inform you that your manuscript has been deemed suitable for publication in PLOS ONE. Congratulations! Your manuscript is now with our production department. 

Kind regards, 

on behalf of

Dr. Matt A Price 

Academic Editor

PLOS ONE